# Hyperspectral Imaging for Mobile Robot Navigation

**DOI:** 10.3390/s23010383

**Published:** 2022-12-29

**Authors:** Kacper Jakubczyk, Barbara Siemiątkowska, Rafał Więckowski, Jerzy Rapcewicz

**Affiliations:** 1Institute of Automatic Control and Robotics, Warsaw University of Technology, 02-525 Warsaw, Poland; 2Łukasiewicz Research Network—Industrial Research Institute for Automation and Measurements PIAP, 02-486 Warsaw, Poland

**Keywords:** mapping, data fusion, path planning, hyperspectral camera

## Abstract

The article presents the application of a hyperspectral camera in mobile robot navigation. Hyperspectral cameras are imaging systems that can capture a wide range of electromagnetic spectra. This feature allows them to detect a broader range of colors and features than traditional cameras and to perceive the environment more accurately. Several surface types, such as mud, can be challenging to detect using an RGB camera. In our system, the hyperspectral camera is used for ground recognition (e.g., grass, bumpy road, asphalt). Traditional global path planning methods take the shortest path length as the optimization objective. We propose an improved A* algorithm to generate the collision-free path. Semantic information makes it possible to plan a feasible and safe path in a complex off-road environment, taking traveling time as the optimization objective. We presented the results of the experiments for data collected in a natural environment. An important novelty of this paper is using a modified nearest neighbor method for hyperspectral data analysis and then using the data for path planning tasks in the same work. Using the nearest neighbor method allows us to adjust the robotic system much faster than using neural networks. As our system is continuously evolving, we intend to examine the performance of the vehicle on various road surfaces, which is why we sought to create a classification system that does not require a prolonged learning process. In our paper, we aimed to demonstrate that the incorporation of a hyperspectral camera can not only enhance route planning but also aid in the determination of parameters such as speed and acceleration.

## 1. Introduction

Over the last decades, we have observed that the market for autonomous cars and transport within industrial plants is developing in many countries.

Research on this type of autonomy system at the Łukasiewicz Research Network–Industrial Research Institute for Automation and Measurements (PIAP) has been conducted since the launch of the project in 2018, called ATENA.

Autonomous systems for terrain UGV (unmanned ground vehicle) platforms with the following leader function were implemented in the field of scientific research and development work for the defense and security founded by the National Center for Research and Development in Poland under the program Future Technologies for Defense in the Competition of Young Scientists. The ATENA project was about developing an autonomous system for unmanned ground vehicles operating in rough, unknown terrains. The ATENA system builds the model of an environment based on the point cloud. This system was tested in the field using two offroad cars modified by the Łukasiewicz-PIAP by adding the Łukasiewicz-PIAP drive-by-wire system, making them UGVs. These cars are shown in Figure 1.

The traversability estimation algorithm used in ATENA transforms the point cloud to a 2.5D grid-based map. A sample of a point cloud from 3D is shown in Figure 2. The main problems occur when the ATENA system is planning clear paths in the field covered by vegetation.

An autonomous system based on the geometry rules is vulnerable to the wrong classification of single vegetation as a stiff obstacle that cannot be passed. This problem is well-known in the field of autonomous driving in rough terrain and could be one of the crucial problems to solve in the field of special and combat use of UGVs. An example of unstructured terrain is shown in Figure 3.

Autonomous systems for a particular use are one of the most innovative and essential aspects of today’s usage of the UGV in modern warfare because UGVs can keep soldiers out of the danger zone. Good maneuverability in rough terrain is the key to survivability on the war field. The autonomous systems must be equipped with the traversability estimation algorithm to have the ability to move in rough terrain. The scope of the work is related to hyperspectral imaging for UGV use and is about adding this functionality to the autonomous systems of Łukasiewicz—PIAP UGVs, which will give a market advantage in this field. Hyperspectral imaging can increase the abilities of UGVs to better plan a safe path in rough terrain. Based on this sensor, it is possible to classify the materials of the terrain ahead of the UGV. This information can be used for modifying the cost of passing the terrain. This approach can be widened to even classify the obstacles in the field as being traversable. Humans do this classification automatically, e.g., when a driver drives a car on the ground road, he automatically decides that lonely high vegetation can be passed. Usually, the driver keeps driving on the road even if the path over the grass is shorter, and the autonomous algorithm must reproduce this behavior. This paper describes a method that covers this behavior in path planning based on traversability estimation with the use of hyperspectral imaging.

### ATENA Navigation System

The ATENA vehicle is equipped with a set of sensors: three LIDAR sensors—Velodyne VLP-16 (16 layers, 100 m range), 5 Basler acA1920-48gc (50 frames per second at 2.3 MP resolution) cameras and the Xsens MTi-G-710 IMU sensor and hyperspectral camera Cubert Q285. The specs of the hyperspectral camera are shown in Figure 4.

Based on Lidar sensor indication, the ATENA system builds a grid-based map of the environment. This part of our research is described in [1]. In a classic approach, the grids are divided into groups—occupied and free from obstacles. In our system, a semantic label is attached to each free-from-obstacles cell. This label represents the type of ground and is defined based on hyperspectral camera indication.

Then modified A* algorithm is used for collision-free path planning. The algorithm takes into account the type of surface. That allows us to consider various optimization criteria in the path-planning process—fuel consumption, route length, and travel time.

This paper is organized as follows: following the introduction is the section discussing related work. In Section 3, the method of ground recognition is explained; this section contains experimental results illustrating the advantages of our approach. Section 4 presents the influence of hyperspectral imaging in the collision-free path planning method considering the cost of driving over different terrain. The article concludes with a summary and bibliography.

## 2. Related Work

Multi- and hyperspectral cameras are used in numerous fields such as geology, ecology, cartography, agriculture, oceanography, medicine, urban planning, monument conservation, or navigation of mobile robots.

UAV (unmanned aerial vehicle) equipped with LIDAR and a multispectral camera is used by a team from Southampton [2] to identify areas of bare ground and vegetation. Although the use of LIDAR alone gives satisfactory results in terms of accuracy, the use of a multispectral camera also allows for a determination of the normalized difference vegetation index (NDVI) and increases the accuracy of area measurements. All these factors can be determined with an accuracy of 0.1 m. A similar example of assessing the quality of barley vegetation is described in [3]. Researchers in Beijing [4] used data from hyperspectral cameras mounted on UAVs to determine wheat yields (as tons per hectare). The correctness of the estimates with this method was determined at 75%. This model can be used in crop planning and management. The publication [5] describes a method of estimating damage to a tree caused by a beetle using data obtained from a hyperspectral camera attached to a UAV. The use of spectral data and appropriate statistical fit models allowed to obtain the accordance of the proposed model with the empirical one at 80%.

Spectral images of the coral reef are used as a non-invasive method to assess the human impact on climate change. Researchers from the UK [6] have developed a system to combine data from different cameras taking pictures of coral reefs (by divers, robots, or airborne UAVs) and obtaining spectral and photogrammetric models to assess the condition of the coral reef using a single shot.

Environmental sciences use hyperspectral cameras mounted on a drone to detect plastic waste floating on the water surface [7]. Researchers present a system for the automatic detection of floating plastic waste based on a random forest classifier. The results strongly depend on the flight altitude of the UAV, and while they are satisfactory in terms of accuracy, due to the detection of many samples as false positives, the issue requires further work.

Multispectral cameras are used to examine the degree of damage to city streets and sidewalks [8]. The UGV is equipped with a set of sensors and cameras. The collected data are then processed to determine whether the pavement requires just maintenance, refurbishment, or reconstruction. This helps city planners to plan road repairs more efficiently. Another example from the field of urban planning is a method of classifying urban materials based on spectral data and textures [9]. This makes it possible to distinguish between similar materials, e.g., concrete, asphalt, or sandstone. Researchers use random forest and support vector machines for classification.

In the field of defense, multispectral images taken by UAVs are used to find landmines and unexploded ordnance [10]. This increases the safety of sappers since standard techniques for finding buried explosives are based on techniques from the mid-twentieth century.

In localization tasks, hyperspectral cameras are used, i.e., for simultaneous quick mapping of terrain and location [11]. The described system consists of multi- and hyperspectral cameras, inertial navigation, and GPS. It can be mounted on an airplane or UAV. The system takes spectral photographs while tracking its position, which significantly facilitates locating the areas of the created map. The application of this system is assessing the condition of forests, damage to green areas after fires, damage caused by insects, and many others. The publication focuses mainly on the presentation of the system, which is supposed to be fast, cheap, and easy to build. A similar system, but described more in terms of algorithms, can be found in [12]. Another example from the localization area is [13]. Researchers addressed the problem of visual odometry in relation to the stereoscopy of multispectral images. A significant problem is the difference in the modalities of the two images. It is proposed to track the features in each image separately and simultaneously and then estimate the motion using the feature parallax. The Kalman filter also makes the system immune to noise resulting from insufficient image quality.

The quality and accuracy of navigation of autonomous vehicles must be ensured regardless of weather conditions. Therefore, the system described in this publication must consider interference from changing lighting (time of day, shadows, etc.) and humidity (e.g., whether wet grass will be as easy to distinguish from the asphalt as dry grass).

In the above-cited publication [3], the researchers set themselves the goal of determining the quality of plant vegetation. They determined that the optics of the hyperspectral camera has an enormous impact on the quality of the measurement, as has the pixel’s location in the image. However, this problem can be ruled out by calibration. In terms of the influence of the environment on the obtained spectra, they distinguished sources of interference: variable insolation, noise from the soil background, plant structure, or shadows.

When using neural networks or graphs to classify materials used in urban environments, much attention must be paid to the quality of the datasets. The authors from [9] did so, pointing out that many of the available databases contain images taken in stable laboratory environments. For better classification, it would be desirable to have data with different levels of exposure or insolation.

In the case of using hyperspectral measurements in the aquatic environment, the authors [6] only mention that for examining a coral reef from the air, the influence of the atmosphere must be considered, unlike for underwater camera shots. However, nothing is mentioned about the impact on the spectra of water alone.

The other works cited above were carried out in stable, favorable conditions, none mentioning the influence of water and moisture on measurements from hyperspectral cameras.

Interesting observations are presented in the work of Andreou et al. [14]. They studied the condition of the asphalt using a multispectral camera. They divided the condition of the asphalt into five categories, ranging from good conditions to poor conditions. They observed that wet asphalt has a spectral response similar to the asphalt under good conditions. Similarly, asphalt with soil on the surface has a spectral response similar to the asphalt under poor conditions.

Knaeps et al. [15] created a database of photos and spectrograms of plastic garbage to train algorithms for detecting pollutants in aquatic environments. The base contains items such as bottles, cups, and plastic packaging that can be wet, dry, full of water, or empty. Spectral responses for selected items are also presented. However, the usefulness of this base in the case of the work described in this publication is insufficient because plastic is not a natural driving surface for an autonomous vehicle.

### 2.1. Spectra Matching and Processing

According to Pozo et al. [16], using a hyperspectral camera requires a few adjustments before the actual radiometric calibration. Those adjustments are background errors and vignetting removal (reducing differences in brightness between the edges and the center of the image). The background can be removed by measuring the dark current, i.e., the camera’s response in the absence of light. The vignetting effect can be reduced by finding tangential and radial distortion corrections. Radiometric calibration can be performed in a laboratory using electromagnetic wave sources of a selected frequency. However, this method is time-consuming and of little use for the cameras currently used on the market. Therefore, in situ radiometric calibration can be performed using canvases of specific colors (shades of grey, red, blue, etc.). The canvases are photographed with a hyperspectral camera to acquire new correction factors for the camera model obtained through laboratory tests. A similar procedure for radiometric calibration is provided in [17].

The processing of spectra from hyperspectral cameras can be carried out in several ways: image spectroscopy (matching pixel spectra with reference spectra), image analysis (with a reduction in multidimensionality to reduce the computational cost, such images are subjected to typical machine learning algorithms in image analysis, such as SVM or CNN) [18]. Researchers at [19] strongly emphasize dealing with multidimensional data. They describe the problem of classifying non-linearly separable data using SVM. The main conclusions from this work are the problem of the low applicability of hyperspectral data to supervised classifiers and the necessity and difficulty of combining data in the spatial domain with the spectral domain.

In order to reduce the computational cost of spectra analysis, the selection of the most informative spectra channels can be used. A proposal for such a method is presented in [20].

In the works cited in the subsection above, many statistical or AI classifiers are also used. For instance, in [14], thresholding, unsupervised classification iterative self-organizing data analysis technique (IsoData), supervised classification spectral angle mapper (SAM), mixture-tuned matched filtering and Fisher linear discriminant are used for the asphalt classification. Spectral angle mapping was used in [10] as well. Torti et al. Cortesi et al. [7] use a random-forest classifier to detect whether a given object floating on a river surface is plastic. Random forest, support vector machine classifiers, and a histogram-based gradient boosting classification tree are used to detect urban material such as asphalt, conglomerate, or sandstone [9].

### 2.2. Traversabilty Estimation

In rough terrain, the environment can rarely be divided into obstacles and free space [21,22]. In the case of UGV, the terrain will have varying degrees of traversability depending on the terrain’s characteristics and the ability of the UGV to traverse the terrain. Driving over rough terrain autonomously is challenging because it is difficult to correctly classify the terrain and quantify its traversability [23].

Traditional path-planning algorithms rely on a model of the world consisting of obstacles and accessible space. The goal of planning is usually to find paths that minimize the distance to travel [24]. Therefore, using classical algorithms for path planning in rough terrain is challenging. For this reason, off-road path planning is typically performed using world models. The world model captures more information than just obstacles/clearance, such as 3D elevation maps, maps with pre-calculated traversability results, segmented maps with terrain types or similar information, or combinations of these. Traversability mapping methods fall into two main categories, geometry-based and appearance-based. More information about traversability, in general, can be found in [25].

For instance, the modified Hybrid A* [26] algorithm has been enhanced with route planning, optimizing the distance to the endpoint and minimizing the cost of travel in the form of road traversability. In experiments, the proposed method was successfully applied to autonomous driving for a distance of up to 270 min in rough terrain. The proposed method provides more traversal paths than the existing Hybrid A* method.

Typically, the traversability estimation is computed using only non-empty cells in the environment, while empty cells (e.g., due to occlusion) are just ignored. As a result, these approaches can only provide traversability maps where terrain data are available. The authors of [27] tackled this problem by proposing, given incomplete terrain data, to make an initial prediction assuming that the terrain is rigid, using a learned kernel function. This initial estimation is then refined to account for the effects of potential land deformation using a near-to-far learning method based on the Gaussian process of multi-regression.

The work [28] presents a standard local and global methodology for planning the construction of continuous cost maps using LIDAR based on the representation of the environment’s traversability. Two approaches are being explored. The first statistical approach calculates the land cost directly from the point cloud. The second approach, based on learning, predicts the IMU response solely from geometric point cloud data using a 2D-Convolutional-LSTM neural network. This allows estimating the cost of the route without directly driving over it.

The analysis of images from cameras using deep learning [29] is also used to study traversability. Pre-trained networks are used that quickly adapt to new conditions.

### 2.3. HSI for Path Planning Tasks

Specifically, in wheeled mobile robots, HSI extends the robot’s ability to acquire information about its surroundings. For example, the authors of [30] show the use of HSI in labeling training data for convolutional neural networks analyzing RGB data, and “shallow” neural networks perform the matching of the spectra from the HS camera. However, in this work, the authors do not mention anything about the specific use of HSI other than better recognition of the area around the robot. The problem they solve is the difficulty of labeling training data for AI algorithms. Similarly, references [31,32,33,34,35] are about environmental classification, not navigation.

The above review shows that there are no works in which the authors would focus on improving the traversability estimation in a wheeled mobile robot using HSI and would demonstrate the effectiveness of this method.

## 3. The Methodology

Our approach uses the supervised learning method to classify the ground based on hyperspectral camera images.

The algorithm consists of the following steps:Data assembling,Data reduction,Normalization and choosing the metric,Creating models of the classes (learning phase).

### 3.1. Data Assembling

The hyperspectral images can be represented as a set of triples:(1)$$\left\{\left(x_{ij},y_{ij},\lambda_{ij}\right)\right\}$$
where *x* and *y* represent 2D spatial dimensions of the scene, and λ represents the 2D spectral dimension. The parameter $\lambda_{ij}$ can be used during the segmentation process.

Figure 5 presents the result of potted plant image segmentation based on spectral responses. The image is represented as a grid of cells, and each area’s average value of λ is computed. It is assumed that two cells belong to the same class if the corresponding values of parameters λ are near each other.

Data for traversability estimation were collected as the robot traveled in different light and weather conditions. The vehicle moved on asphalt, a bumpy road, a forest road, and grass. Figure 6 shows images of each type of road.

The photos are divided into the square cells shown in Figure 6 (red grid). A spectral distribution is created for each cell, and a label (class) is assigned.

### 3.2. Data Reduction

The distributions of the spectral response of areas belonging to the same class and similar values are averaged using the kernel density estimation (KDE) technique [36].

Figure presents the idea behind the method. On the left are images obtained from the hyperspectral camera. The squares indicate selected road parts for which wavelength distributions were created. The figures on the right show the spectral distributions. The solid line marks the result of averaging. In our approach, spectral distribution (λ) is treated as a feature vector and is used during classification.

To avoid the influence of lighting conditions, the values of λ have been normalized according to the formula:(2)$$r_{i}^{N}=\frac{r_{i}}{r_{max}-r_{min}}$$
where $r_{i}$—intensity of *i*th wavelength.

During the experiments, it turned out that the same type of area (grass) has different values of λ for different seasons (Figure 7). Therefore, some classes are represented by several vectors of features.

To classify an unknown area, we need to find the nearest area with a known class membership [37]. For this purpose, we have to define a metric to compare the distance between spectral distributions. In our algorithm, we adopted methods of comparing histograms [38].

### 3.3. Choosing the Proper Distance Function

The choice of the appropriate distance function significantly affects the classification result. We adopted the methods that are used to compare histograms: the correlation between histograms (Equation (Equation 3)), the product of histograms (Equation (Equation 5)), and evaluating distances using the chi-square method (Equation (Equation 6)). $x_{i}$ and $y_{i}$ are the corresponding values of the histogram pair (X,Y). It is assumed that $x_{i}$ and $y_{i}$ are the corresponding values of the histogram pair (X,Y).
(3)$$d_{c}\left(X,Y\right)=\frac{\sum_{i=1}^{n}\left(x_{i}-\hat{x}\right)\left(y_{i}-\hat{y}\right)}{\sqrt{\sum_{i=1}^{n}{\left(x_{i}-\hat{x}\right)}^{2}{\left(y_{i}-\hat{y}\right)}^{2}}}$$
where:(4)$$\begin{array}{ccc} \hat{x} & = & \frac{\sum_{i=1}^{n}x_{i}}{n},\\ \hat{y} & = & \frac{\sum_{i=1}^{n}y_{i}}{n} \end{array}$$
$d_{c}\left(X,Y\right)$)—correlation between histograms.
(5)$$d\left(X,Y\right)=\sum_{i=1}^{n}min\left(x_{i},y_{i}\right)$$
d(X,Y)—product of histograms
(6)$$d_{c2}\left(X,Y\right)=\sum_{i=1}^{n}\frac{{\left(x_{i}-y_{i}\right)}^{2}}{\left(x_{i}+y_{i}\right)}$$
$d_{c2}\left(X,Y\right)$—chi-square methods.

The distance between two different classes should be as large as possible. The metric values for elements belonging to the same class should be close to zero. The experiments show that the best results were obtained for the chi-square function.

### 3.4. Classification Phase

In our approach, the nearest-neighbor supervised learning algorithm is used. The examples in the dataset have labels assigned to them, and their classes are known (Section 3.2). Figure 8 shows the idea behind the method. The algorithm consists of the following steps: (1) calculating the distance between a test example and dataset examples, and (2) finding the minimum distance and corresponding class label. The label is the result of the classification.

### 3.5. Experimental Results

Classification accuracy was determined by adopting a chi-square metric. Figure 9 shows the confusion matrix (in percent). The matrix compares the actual class values with those predicted by the learning model. Vegetated terrain is not confused with other terrains.

After classifying cells of the area, semantic information should be taken into account. For example, if a cell of an image is classified as grass and is surrounded by cells classified as a road, the classification result is changed. This method resembles the method of n-nearest neighbors [37].

Figure 10 shows the result of the classification. The gray color shows parts of the asphalt road, and the green one shows the grass. Context is taken into account in the classification process, i.e., if a single piece is surrounded by fragments belonging to another class, the value is changed.

## 4. Application of Surface Recognition in Collision-Free Path Planning

In this article, we suggest using the diffusion method for path planning [1,39]. The main advantage of the approach is that we can easily consider the cost of driving over different terrains.

The map of the environment is represented as a grid of cells (array map). Obstacle-free cells represent the possible robot positions (states). Two states are distinguished: the initial robot position ($c_{R}$) and the goal position ($c_{G}$). In classical path planning systems, we divide cells into two classes: free from obstacles and occupied. For 2.5D maps, class membership is determined by thresholding. A cell is classified as occupied if the observed height exceeds a certain threshold and is free of obstacles otherwise. Figure 11 presents the map built by the ATENA navigation system [1]. Red fragments represent areas occupied by obstacles, green—free of obstacles, white—unexplored. The algorithm consists of the following steps:A diffusion map (array *v*) is initialized in the path planning algorithm’s first step. The big value is assigned to the cell, representing the goal position ($c_{G}$), and the values 0.0 are attached to other cells.For each unoccupied cell ($map_{ij}$), the value ($v_{ij}$) is calculated according to the formula:
(7)$$v_{ij}=max_{v_{ekl}\in N_{ij}}\left(v_{ekl}-dist\left(c_{kl},c_{ij}\right)\right)$$
where $N_{ij}$—neighborhood of the cell ij, dist—distance between cells. This process continues until stability is established.The path (list of cells) is generated during the next step. The first cell represents the initial robot’s position. The next one is indicated by the neighbor of ($c_{R}$) with a maximum value of $v_{kl}$. The process continues until the cell with the maximum function value is reached. Figure 11b represents the path generated by the algorithm. A black line indicates the planned path. R—the robot’s initial position, G—the target’s location.

In the case of outdoor robots, we look for the path with the shortest travel time. The travel time (robot speed) can be related to the type of ground, distance to obstacles, etc. To solve this problem, we have to modify the map of the environment and the diffusion method. The map of the environment represents the cost function. The value depends on the type of ground detected, thanks to hyperspectral imaging. In the case of a perfectly smooth terrain (asphalt), the value of this parameter equals 0.0. In the case of uneven terrain, the value is increased proportionally to the increase in movement cost (time).

Figure 12 presents the map if the type of ground is considered. Red fragments represent areas occupied by obstacles, green—grass, gray—asphalt, white—unexplored.

Equation (Equation 7) is modified as follows:(8)$$v_{ij}=max_{v_{ekl}\in N_{ij}}\left(v_{ekl}-dist\left(c_{ekl},c_{aij}\right)-map_{kl}\right)$$

Figure 12b represents the path generated by the algorithm. A black line indicates the planned path. R—the robot’s initial position, G—the target’s location.

The path’s start and end are exactly the same as in Figure 11, but because the cost of travel for different terrains is taken into account, the paths differ significantly. Most of the second path runs on asphalt.

### Discussion

We want to show the advantages of our system by comparing it with the algorithm presented in [40]. The paper proposes a hierarchical path-planning method for a wheeled robot. It is assumed that the vehicle moves in a partially known uneven terrain, and the 3D map stores elevation data, such as the terrain’s slope, step, and unevenness. The A* algorithm is used for global path planning. The Q-learning method is employed to avoid locally detected obstacles.

In our opinion, geometric information about the surface is insufficient when the robot moves on difficult, rugged, uneven terrain. An area defined as impassable based on the elevation map (for example, high tufts of grass) may be accessible to the robot. Without knowledge of the ground structure, it is difficult to determine whether the robot can drive up a given hill. A hill covered by asphalt is more accessible than a hill covered by mud; both hills could have the same geometry, but the cost of passing them through is different. Additionally, hyperspectral imaging can give information that will change the permissible speed of the robot on a given surface. Similarly, in the case of the Q-learning algorithm, it is worth taking into account semantic information about obstacles.

## 5. Conclusions and Future Works

In the paper, we presented the use of a hyperspectral camera in the navigation of a mobile robot. An effective surface classification method has been developed. An important novelty of this paper is the use of a modified method of the nearest neighbor. Thanks to this solution, we can quickly teach the system to add and remove surface classes. This feature is essential when the system is being developed and modified. Unlike neural networks, the algorithm is transparent and non-parametric. Transparency of classification is vital in security systems. In the case of *CNN*, if an image’s context is misleading or confusing, it can cause the model to make incorrect predictions. Experiments carried out in the natural environment have shown the effectiveness of the proposed solution. It has been shown that information about the surface type can be easily included in path planning and determining the permissible speeds of the vehicle. In the future, we plan to expand the system by adding new classes of surface types. We want to use the surface type information in global path planning and in controlling the robot, e.g., determining the allowable values of linear or angular acceleration. Recognition of surfaces with the help of a camera has been described in the literature, but we have not found an article in which information from a hyperspectral camera was used to determine the cost of driving on a given surface and the permissible speeds. Our experience shows that the current price of hyperspectral cameras for a robot moving in a structured environment is not profitable. Using a hyperspectral camera on a robot moving in difficult, uneven terrain can significantly improve the system’s efficiency. It can also increase the safety of the mission. In the current version of the system, the vehicle’s dynamic is taken into account to a minimal extent. The mass of a vehicle, coefficient of friction, and/or torque limitations can make a particular surface impassable for the robot. The corresponding grid-based map cells are marked as occupied by obstacles. The cost of travel depends on the distance from obstacles. In the future, we will adopt the methods described in [41,42] to our path planning algorithm to allow the robot to move with higher velocities and consider the dynamic to a broader extent.

We plan to expand the system by adding a robot self-location module, in which semantic information will be used in addition to metric information.

In the articles discussed in Section 2, the robots used the hyperspectral cameras to carry out specific tasks, such as plant irrigation tests, not in path planning, so the system proposed in this work can be successfully used in the robots mentioned above that already have an HS camera installed for other purposes. Nevertheless, the price of hyperspectral cameras is high, but it is constantly falling. Ten years ago, the price of 3D laser rangefinders was also high, and now they are standard equipment for mobile robots. In the case of a natural, unstructured environment, the use of information obtained from a hyperspectral camera can improve not only the efficiency but also the security of the system.

## Figures and Tables

**Figure 1 sensors-23-00383-f001:**
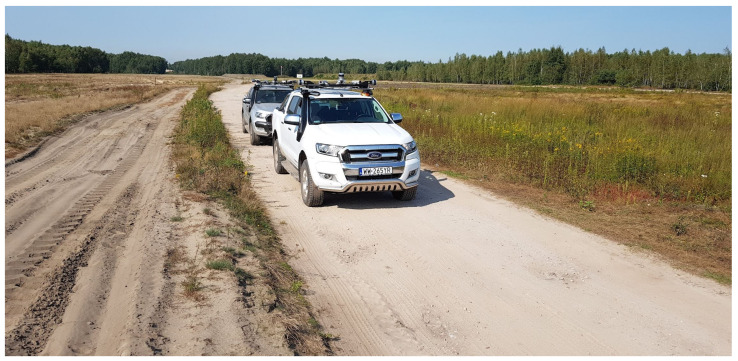
Łukasiewicz-PIAP ATENA demonstrator of technology.

**Figure 2 sensors-23-00383-f002:**
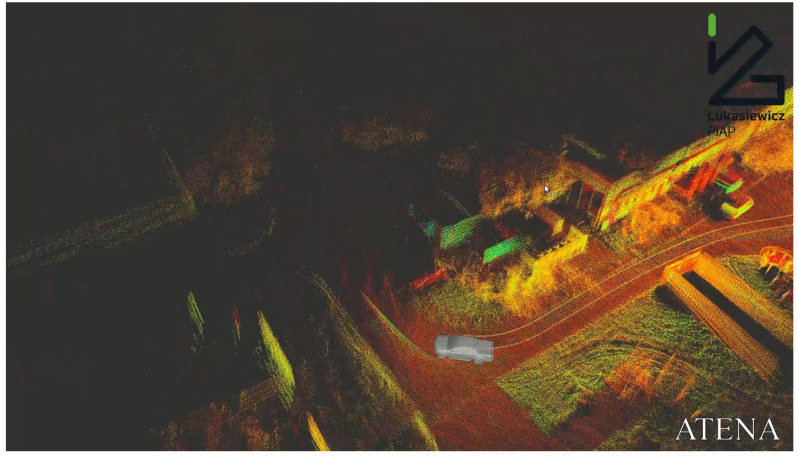
Łukasiewicz-PIAP ATENA demonstrator of technology in obtaining 3D data.

**Figure 3 sensors-23-00383-f003:**
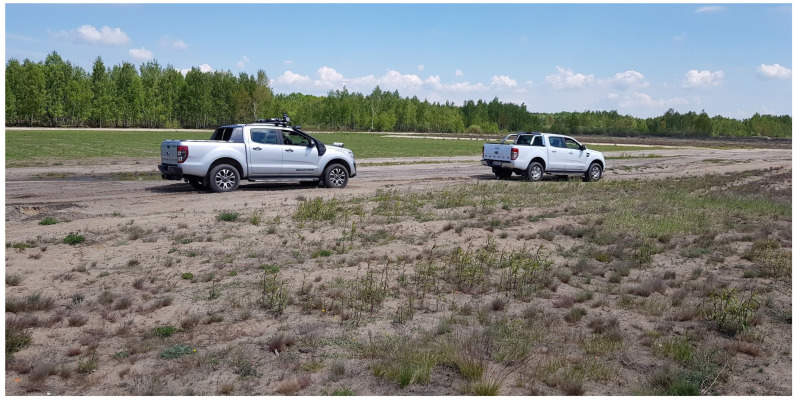
Łukasiewicz-PIAP ATENA demonstrator of technology in unstructured terrain.

**Figure 4 sensors-23-00383-f004:**
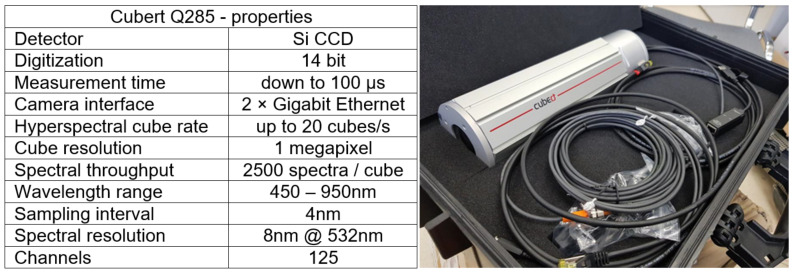
The image and table with properties of the hyperspectral camera Cubert Q285.

**Figure 5 sensors-23-00383-f005:**
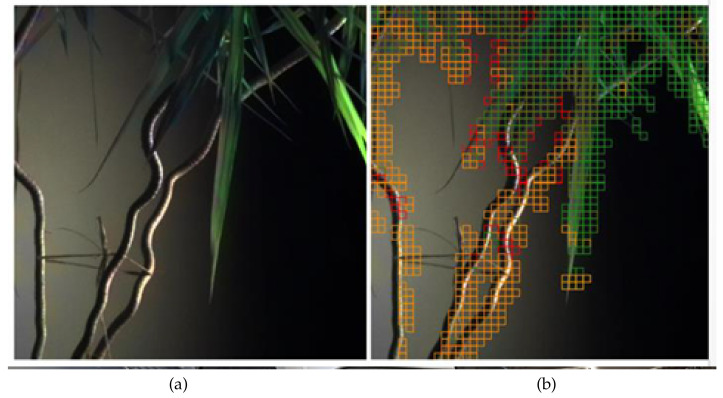
The image taken by a Cubert Q285 camera (**a**), and the result of segmentation (**b**).

**Figure 6 sensors-23-00383-f006:**
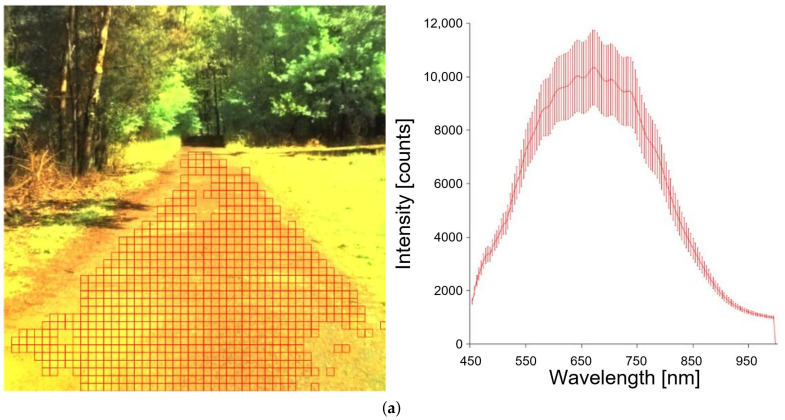
Different types of roads and corresponding spectral response: (**a**) dirt road, (**b**) forest road (land), (**c**) bumpy/mixed road (with ruts).

**Figure 7 sensors-23-00383-f007:**
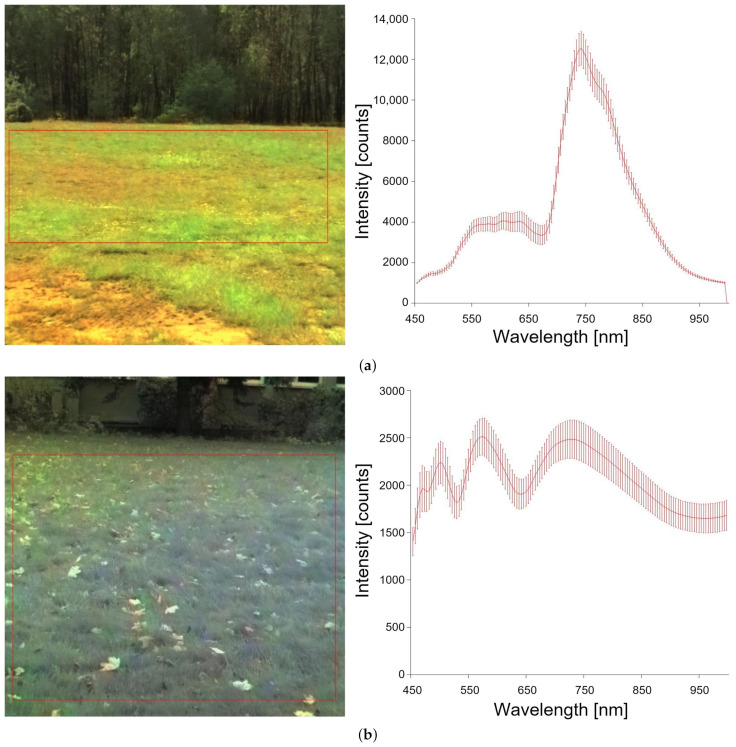
Grass at a different time of year: (**a**) grass in May, (**b**) grass in October.

**Figure 8 sensors-23-00383-f008:**
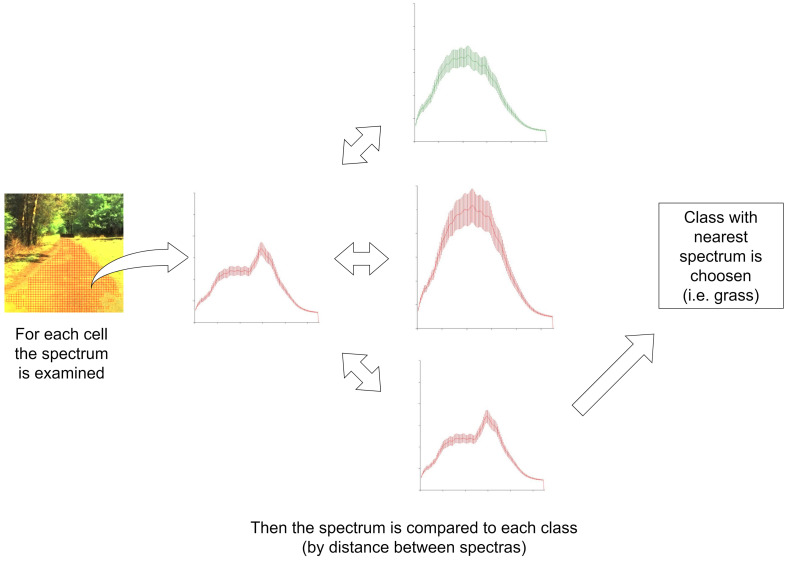
Classification diagram.

**Figure 9 sensors-23-00383-f009:**
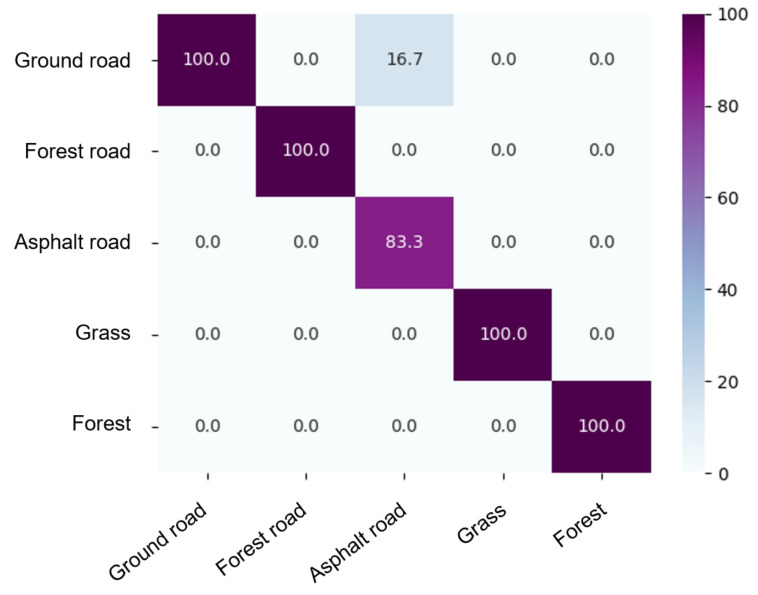
Confusion matrix for terrain classification matrix (%).

**Figure 10 sensors-23-00383-f010:**
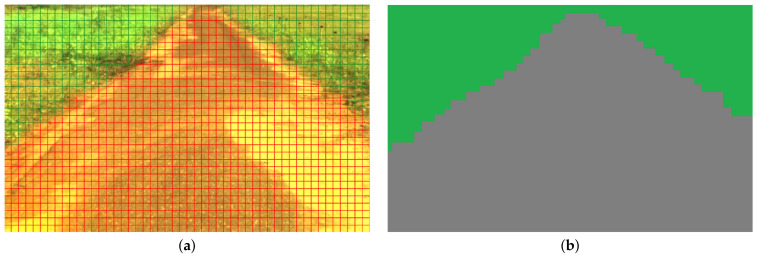
The segmentation results, (**a**) original image, (**b**) image after segmentation.

**Figure 11 sensors-23-00383-f011:**
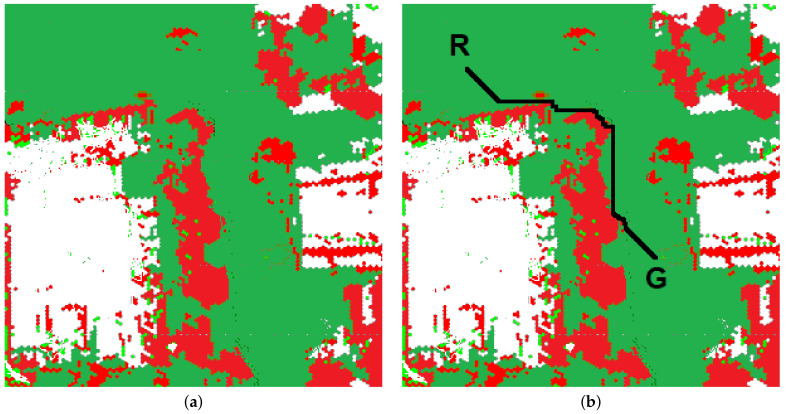
The map of the environment: (**a**) the map, (**b**) the path.

**Figure 12 sensors-23-00383-f012:**
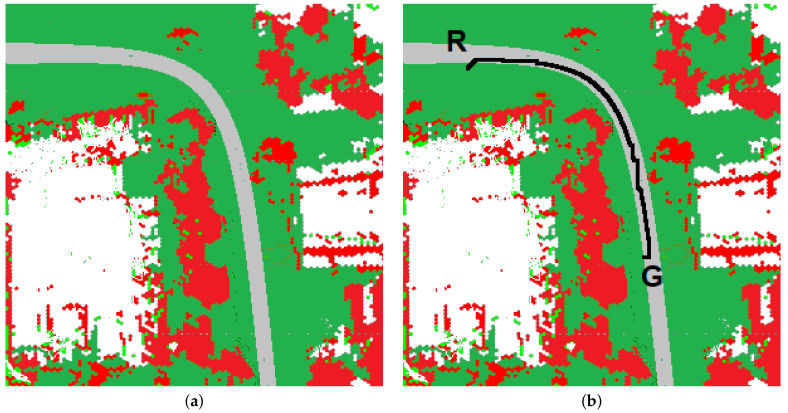
The map of the environment if the type of ground is recognized: (**a**) the map, (**b**) the path.

## Data Availability

The source code and data used to support the findings of this study are available from the corresponding author upon request.

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
