# Peer review of "Hyperspectral Imaging for Mobile Robot Navigation"

_sensors, 2022, doi:10.3390/s23010383_

Round 1

Reviewer 1 Report

In this article, authors have suggest classifying the type of ground using a hyperspectral camera and comparing spectral distribution to help mobile robots better navigate their environment. The evaluation of traversability is a crucial component of the navigation system. It is frequently created using a height map. In the real world, information on the type of terrain is crucial in addition to data about obstacle height ( e.g. grass, bumpy road, asphalt). We discussed the experimentation findings for information gathered in a natural setting. Authors have demonstrated how well the path planning method can employ information about the sort of region. Before final publication, this work must address major  revisions/concerns.

1. Revise the abstract. Need to write the abstract in professional manner.

2. What is novelty of the work. Please underscore the scientific value added/contributions of your paper in your abstract and introduction and address your debate shortly in the abstract.

3. A good article should include, (1) originality, new perspectives, or insights; (2) international interest; and (3) relevance for governance, policy or practical perspective.

4. The work is devoted to an actual scientific and applied problem, performed by correct modern methods and the results are not in doubt. But the presentation and discussion of the results, as well as the conclusions, need to be improved.

5. Figure 1 and Figure 2 are not cited in the text. Why these figures are provided.

6. Abbreviations should be defined at first mention and used consistently thereafter. Check and correct.

7. Figure 5 is not clear. Improve it.

8. Figure 7. Average distances between classes. Why are distances presented in form of confusion matrix?

9. Compare the results with other state-of-the-art methods.

Author Response

Thank you very much for your valuable comments. We have tried to take all the comments into account. Below are the detailed answers.

  1. Revise the abstract. Need to write the abstract in professional manner. - The abstract has been rewritten.
  2. What is novelty of the work. Please underscore the scientific value added/contributions of your paper in your abstract and introduction and address your debate shortly in the abstract.  The novelty of the work was underline in the abstract and introduction.
  3. A good article should include, (1) originality, new perspectives, or insights; (2) international interest; and (3) relevance for governance, policy or practical perspective. We bordered the conclusion and practical perspective were added.
  4. The work is devoted to an actual scientific and applied problem, performed by correct modern methods and the results are not in doubt. But the presentation and discussion of the results, as well as the conclusions, need to be improved. The article has been improved.
  5. Figure 1 and Figure 2 are not cited in the text. Why these figures are provided.  Figures were described.
  6. Abbreviations should be defined at first mention and used consistently thereafter. Check and correct. Abbreviations have been defined.
  7. Figure 5 is not clear. Improve it. Figure 5 was improved
  8. Figure 7. Average distances between classes. Why are distances presented in form of confusion matrix?  Figure 7 was removed
  9. Compare the results with other state-of-the-art methods.  The classification method was compared with neural network approach.

Reviewer 2 Report

1. Too many related applications of the spectral image are shown in section 2, which aren’t very associated with this topic closely. Please introduce more works about solving the problems of hyperspectral sensoring and processing.

2. The instrumental parameters of the hyperspectral camera aren’t introduced in this paper. And the advantage of this target identification compared with deep learning doesn’t show enough. Economic analysis should be considered for this application.

3. The path planning method doesn’t tie closely with the hyperspectral imaging technology for pathfinding.

4. Using a hyperspectral camera to indentify botany and terrain isn’t a new method.

5. The presented path planning method isn’t novel. And the dynamic of the robot should be considered in the path planning method.

Author Response

Thank you very much for your valuable comments. We have tried to take all the comments into account. Below are the detailed answers.

  1. Too many related applications of the spectral image are shown in section 2, which aren’t very associated with this topic closely. Please introduce more works about solving the problems of hyperspectral sensoring and processing Section 2 was rewrite
  2. The instrumental parameters of the hyperspectral camera aren’t introduced in this paper. And the advantage of this target identification compared with deep learning doesn’t show enough. Economic analysis should be considered for this application. The parameters were added
  3. The path planning method doesn’t tie closely with the hyperspectral imaging technology for pathfinding.

The cost map (used by path planning module) is built based on data about type of surface. The type of surface is defined using hyperspectral imaging.

  1. The presented path planning method isn’t novel. And the dynamic of the robot should be considered in the path planning method.

In the current system version, the vehicles move slowly, and the vehicle's dynamic is taken into account to a minimal extent. The mass of a vehicle, coefficient of friction,  and/or torque limitations can make a particular surface impassable for the robot, and the corresponding cells are labelled as occupied  In the future, we will modify our path planning algorithm to allow the robot to move with higher velocities and consider the dynamic to a broader extent. 

Reviewer 3 Report

sensors-2095457

In this article, the authors presented a hyperspectral camera and compared spectral distribution to classify the type of ground to improve path-finding tasks for mobile robots. An essential element of the navigation system is the assessment of traversability. The author’s work is interesting, but still, there are some limitations. Some of them are as follows:

1.      There are some grammatical mistakes and typos that need to be corrected with detail proofread.

2.      The abstract needs to be written better. The authors are suggested to look at some quality abstracts and then rewrite the abstract.

3.      Figure 5 and Figure 6 are very blurred; please add original figures. Also, modify the text of the figure.

4.      The font size and format of the figure’s text must be consistent with the paper body text.

5.      Similarly, Figure 8 seems very blur. Also, it would be better to add some more descriptions to the captions.

6.      Additionally, add a full stop at the end of every caption of the figures.

7.      The caption of “Figure 9. Error matrix (%)” is not clear. Please add more description.

8.      The current results seem like they need to be improved. I suggest doing some more experiments and adding some quality results.

9.      The technical depth of the paper could be improved too.

10.   Comparison of the proposed work with other’s work needs to be included. Therefore, accurately comparing their model to other robot navigation approaches is preferable. If there is no comparison, how can the novel reader understand that this model is novel or effective? Therefore, I highly advise comparing your model to the most recent models, particularly those from the last two years.

11.   The conclusion is poorly written. The authors need to summarize the approaches that have been adopted, the findings that have been identified, and the problems that still exist, and then make some suggestions or ideas for future study.

Author Response

Thank you very much for your valuable comments. We have tried to take all the comments into account. Below are the detailed answers.

  1. There are some grammatical mistakes and typos that need to be corrected with detail proofread.

The paper was corrected.

  1. The abstract needs to be written better. The authors are suggested to look at some quality abstracts and then rewrite the abstract. The abstract was rewritten
  2. Figure 5 and Figure 6 are very blurred; please add original figures. Also, modify the text of the figure. The figure were modified
  3. The font size and format of the figure’s text must be consistent with the paper body text. It has been modified
  4. Similarly, Figure 8 seems very blur. Also, it would be better to add some more descriptions to the captions. Figure 8 was corrected.
  5. Additionally, add a full stop at the end of every caption of the figures. The full stops were added
  6. The caption of “Figure 9. Error matrix (%)” is not clear. Please add more description. .  Figure 8 was corrected.
  7. The technical depth of the paper could be improved too. . The paper was improved.
  8. Comparison of the proposed work with other’s work needs to be included. Therefore, accurately comparing their model to other robot navigation approaches is preferable. If there is no comparison, how can the novel reader understand that this model is novel or effective?

In the paper, we do not present a new navigation algorithm. The scheme of navigation systems in most algorithms is similar. From our point of view, it is important that thanks to the use of a hyperspectral camera, an area covered with tall grass can be treated as passable and covered with mud as impassable. The advantage of our method is a transparent and flexible classification system.

  1. The conclusion is poorly written. The authors need to summarize the approaches that have been adopted, the findings that have been identified, and the problems that still exist, and then make some suggestions or ideas for future study.

The conclusion has been rewritten.

Round 2

Reviewer 1 Report

Authors have addressed all revisions/comments. Now it is acceptable in its present form.

Author Response

We would like to extend our heartfelt thanks for your review.

Reviewer 2 Report

The novelty of this paper should be reconsidered still. The authors just add a hyperspectral camera to increase the ability of navigation. It isn’t a new method. Though the detection capacity is better, the cost of the whole system is expensive. So, please compare this work to others to highlight the creativity.

Author Response

We would like to extend our heartfelt thanks for your review. We have added a comparison of our approach and the method described in a recently published article.

Reviewer 3 Report

NA

Author Response

(The authors gave the same response as above.)
